# Identification of the Regulatory Role of *SlWRKYs* in Tomato Defense against *Meloidogyne incognita*


**DOI:** 10.3390/plants12132416

**Published:** 2023-06-22

**Authors:** Weidan Nie, Lili Liu, Yinxia Chen, Mingyin Luo, Chenghao Feng, Chaonan Wang, Zhongmin Yang, Chong Du

**Affiliations:** College of Horticulture, Xinjiang Agricultural University, Urumqi 830052, China; 13779334686@163.com (W.N.); liulili4477@163.com (L.L.); cyx61222@163.com (Y.C.); m18253133966@163.com (M.L.); fch20220515@163.com (C.F.); wcn0107@126.com (C.W.); yangzhongmin161220@126.com (Z.Y.)

**Keywords:** root-knot nematode, tomato, disease resistance, WRKY, VIGS

## Abstract

Root-knot nematode (RKN) infections are among the most serious soil-borne diseases in the world, and tomato is a common host of RKNs. WRKY transcription factors are involved in complex, diverse biological processes in plants. In a previous study, a resistant variety, LA3858 (*Mi-3*/*Mi-3*), was treated at different soil temperatures before RNA-seq, and six differentially expressed genes (DEGs) encoding WRKY proteins were screened. In this study, cloning and sequencing were used to identify six target DEGs encoding SlWRKY1, SlWRKY13, SlWRKY30, SlWRKY41, SlWRKY46, and SlWRKY80. Conserved domain identification and phylogenetic tree analysis showed that SlWRKY1, SlWRKY13, and SlWRKY46 have similar functions and are mainly involved in plant growth and development and abiotic stress responses. SlWRKY30 and SlWRKY41 share high homology, while AtWRKY46 and AtWRKY70, which are highly homologous to SlWRKY80, play an important role in the disease resistance of *A. thaliana*. Considering these findings combined with the high level of *SlWRKY80* expression observed in the roots and leaves of the resistant variety Motelle (*Mi-1*/*Mi-1*) and the continuous upregulation of *SlWRKY80* expression in the roots after inoculation of Motelle with *M. incognita*, it is speculated that *SlWRKY80* plays an important role in the *Mi-1*-mediated disease resistance pathway. Further study revealed that SlWRKY80 is a typical nuclear-localized protein, and a virus-induced gene silencing (VIGS) assay verified that *SlWRKY80* is involved in tomato resistance to RKNs as a positive regulator. SA and JA signals play an important role in *Mi-1*-mediated resistance to RKNs. *SlWRKY80* was able to respond rapidly to treatment with both plant hormones, which indicated that *SlWRKY80* might be involved in disease resistance regulation through various immune pathways.

## 1. Introduction

Root-knot nematodes (*Meloidogyne* spp.) are among the most harmful plant-parasitic nematodes in the world. They can infect more than 5500 host species, including food crops, cash crops, oil crops, vegetables, and fruits [1]. The life cycle of RKNs includes three stages: egg, larva, and adult. In soil, RKNs infect host plants mainly as second-stage juveniles (J2s), which can penetrate the apical epidermis and reproduce in the roots via the intercellular space [2]. At present, there are four species of *Meloidogyne* known to have a serious impact on agricultural production, including *M. incognita*, *M. hapla*, *M. javanica*, and *M. arenaria*. Among these species, *M. incognita* is the most widely distributed [3,4]. *M. incognita* can feed on most vascular plants and induce abnormal cell division around the area where it occurs, resulting in cell enlargement and the formation of giant cells with multiple nuclei to provide nutrition for its own growth and development [5], eventually forming distinct root knots on the root surface, which is a typical symptom of *Meloidogyne* [6]. RKNs have caused significant losses to the world agricultural economy and have become an obstacle to current crop production. According to reports, crop yield reductions caused by RKNs can reach 30–40% in tropical and subtropical regions and, in severe cases, can exceed 70% or even lead to crop failure, resulting in economic losses of hundreds of billions of dollars annually [7,8].

To date, exploiting the *R* gene has been the most effective way to improve crop resistance to disease [9]. Tomato (*S. lycopersicum*) is a conventional host of RKNs and is often attacked by nematodes; the main gene family related to resistance to RKNs in tomato is the *Mi* gene family [10]. Although there are 10 members in this family, these resistance genes all come from wild species, and the incompatibility of distant hybridization makes it difficult to mine the members of this family; hence, only *Mi-1* has been directly applied thus far [11]. Due to the lack of available resistance sources, it is difficult to use other *Mi* genes for tomato disease resistance breeding.

Although *R* genes mediate plant resistance to pathogens, resistance is regulated by a variety of biological factors, including an important class of DNA-binding proteins, the WRKY transcription factors. The conserved sequence “WRKYGQK” at the N-terminus of the DNA binding domain leads to the specific binding of WRKY transcription factors to W-box elements in the promoter region of the target gene, whereby they participate in complex biological pathways [12]. Based on this feature, WRKY proteins can be modified by MAPK cascade activation signal phosphorylation to regulate pattern-triggered immunity (PTI) and effector-triggered immunity (ETI) processes [13]. At the same time, many WRKY proteins are common components of SA (salicylic acid)/JA (jasmonic acid)-mediated plant defense pathways, including the induction of systemic acquired resistance (SAR) and induced systemic resistance (ISR) [14]. In addition, WRKYs can change plant disease resistance by influencing effector proteins to form complexes or directly regulating *R* gene expression [15].

With the continuous exploration of WRKY functions in the field of plant disease resistance, it has been found that some *SlWRKYs* are involved in regulating tomato resistance to RKNs. For example, *SlWRKY72*, *SlWRKY73*, and *SlWRKY74* all actively regulate *Mi-1*-mediated ETI processes opposing RKNs and potato aphids [16]. Previous studies have shown that the overexpression of *SlWRKY45* can decrease the expression levels of marker genes (*PR-1* and *Pin2*) in JA and SA signaling pathways [17]. Further studies have revealed that *SlWRKY45* can specifically bind to the promoter of the JA synthesis gene *SlAOC* and inhibit its expression [18]. Thus, the susceptibility of plants to RKNs is enhanced. *SlWRKY3* and *SlWRKY35* expression is significantly induced at tomato root feeding sites after *M. javanica* infection, and *SlWRKY3* responds positively to SA signals. The overexpression and knockout of *SlWRKY3* indicated that it positively regulates tomato resistance to *M. javanica* [19]. In addition, there are few reports on the involvement of other WRKY proteins in tomato resistance to RKNs. Therefore, it is very important to explore more WRKY proteins and their functions to better understand the mechanism by which WRKYs regulate tomato disease resistance. In this study, six differentially expressed genes (DEGs) encoding SlWRKYs were screened according to preliminary RNA-seq results obtained by using the wild-resistant variety LA3858 (*Mi-3*/*Mi-3*), which shows different susceptibilities at various soil temperatures (SRA:PRJNA793629). Through cloning, structural analysis, and expression pattern evaluation, valuable target *SlWRKYs* were screened for preliminary functional verification by virus-induced gene silencing (VIGS), which provides a solid theoretical basis for tomato resistance improvement in the later stage.

## 2. Results

### 2.1. Homologous Recombinant Cloning of SlWRKYs

After homologous recombinant cloning of the DEGs encoding SlWRKY proteins with the vector *pBI121*, positive clones were selected and verified by PCR using the designed specific primers, and bands with the same size as the target gene were obtained (Figure 1). According to the sequencing test, the results were consistent with those of RNA-seq. BLAST searches against the NCBI database identified a total of six DEGs, *Solyc12g006170.2*, *Solyc04g051540.3*, *Solyc10g009550.3*, *Solyc01g095630.3*, *Solyc08g067340.3,* and *Solyc03g095770.3*, encoding SlWRKY1, SlWRKY13, SlWRKY30, SlWRKY41, SlWRKY46, and SlWRKY80, respectively.

### 2.2. Screening and Identification of Conserved SlWRKYs Domains

MEME was used to analyze the amino acid sequence of *SlWRKYs* and five motifs with well-conserved characteristics were selected according to the *p*-value. There were some differences in the position and quantity of each motif in the amino acid sequences of *SlWRKYs* (Figure 2a). Motifs 1 (29 aa) and 2 (22 aa) were highly conserved in different SlWRKYs. Therefore, it was speculated that motif 1 and motif 2 together constituted conserved regions of WRKY transcription factors. Based on the above results, Clustal 2.1 was used for the sequence alignment of SlWRKYs, and its conserved domain was further evaluated (Figure 2b). The results showed that there were 20 highly conserved amino acid residues in the domain, and WRKYGQK in the WRKY domain was conserved across all SlWRKYs; the number of residues was also the same, indicating that the SlWRKYs belonged to types Ⅱ and Ⅲ. According to the zinc-finger-C2H2/C2HC-type structure, SlWRKY30, SlWRKY41, and SlWRKY80 belonged to type Ⅲ, and SlWRKY1 and SlWRKY13 belonged to type Ⅱc. SlWRKY46 belonged to type Ⅱa.

### 2.3. SlWRKYs Phylogenetic Tree Analysis

PlantTFDB was used to screen WRKY proteins in *Oryza sativa* L. and *A. thaliana* that were highly homologous to the SlWRKYs (Score ranking top 10). Duplicate sequences were removed, and 78 WRKY transcription factors were ultimately obtained. The results of the phylogenetic tree analysis performed in this study (Figure 3) showed that the members to be analyzed were divided into six groups (Ⅰ-Ⅵ), among which only two members were in group Ⅲ. SlWRKY1, SlWRKY13 and SlWRKY46 were distributed in group Ⅱ, group Ⅰ, and group Ⅳ, respectively, but there was no significant difference in the number of members among the three groups. The number of WRKYs in group Ⅴ was highest, with 22 members, including SlWRKY30, SlWRKY41, and SlWRKY80, and SlWRKY30 and SlWRKY41 had relatively high homology. SlWRKY80 showed high homology with AtWRKY70 and AtWRKY54 in *A. thaliana*, which are often involved in regulating plant resistance to biological stress.

### 2.4. Tissue Expression Analysis and Root Expression Pattern of SlWRKYs

The expression levels of *SlWRKYs* in the root, stem, and leaf tissues in the four-true leaf stage of Motelle-resistant variety plants were evaluated by RT–qPCR (Figure 4). The results showed that *SlWRKYs* were mostly expressed in the roots and leaves of the individual plants, especially *SlWRKY30* and *SlWRKY80*. The relative expression values of these two genes in roots were 3.41-fold and 5.89-fold, respectively, and their expression levels were also relatively high in leaves, at 3.83-fold and 4.29-fold, respectively. The relatively high expression levels indicate that *SlWRKY30* and *SlWRKY80* are likely to function in various biological processes of tomato.

The expression patterns of *SlWRKYs* in the roots of the resistant Motelle and susceptible M82 tomato varieties were analyzed at the key time points of three dpi (days post-inoculation) and six dpi (the expression level of DEGs was highest in this stage) after inoculation with *M. incognita* (Figure 5). The results showed that, except for *SlWRKY41*, which was slightly downregulated at three dpi, the other five *SlWRKYs* showed upregulated expression at this stage in the Motelle tomato. *SlWRKY30* and *SlWRKY80* presented the highest upregulation ratios of 3.08-fold and 3.21-fold, respectively. In the M82 tomato, with the exception of *SlWRKY46*, the other five *SlWRKYs* all showed downregulated expression trends in this stage; therefore, the expression levels of *SlWRKY30*, *SlWRKY46*, and *SlWRKY80* showed extremely significant differences in the Motelle and M82 lines. At around six dpi, the expression of *SlWRKY13*, *SlWRKY30*, *SlWRKY41*, and *SlWRKY46* decreased in the Motelle tomato. However, *SlWRKY1* and *SlWRKY80* showed a trend of continuously upregulated expression, and *SlWRKY80* showed a high expression value of 4.56-fold. Once again, the expression levels of *SlWRKY80* showed significant differences between the Motelle and M82 lines at six dpi.

### 2.5. Determination of Endogenous SA and JA Contents in Motelle and M82 Tomato after Inoculation

The levels of endogenous SA and JA accumulated at different stages in Motelle and M82 tomato inoculated with *M. incognita* were measured (Figure 6). The results showed that there was no significant difference in both endogenous SA and JA contents between uninoculated Motelle and M82. After inoculation with *M. incognita*, the SA and JA contents of M82 increased overall with time after inoculation, but the differences at various time points were not significant. After inoculation (starting at six hours post-inoculation, or hpi), the SA and JA contents of Motelle plants at each postinoculation time point were significantly higher than those of M82 plants. Additionally, SA accumulation reached its peak 12 h after inoculation, i.e., 2.96 times higher than that of M82 at the same time point. The content of JA reached its peak at the 6th hour after inoculation, i.e., 3.65 times higher than that of M82. After reaching their peaks, the levels of SA and JA in Motelle began to decline and then stabilize. The above findings further demonstrate that the SA and JA hormone signaling pathways play an important role in the *Mi-1*-mediated resistance pathway in Motelle tomatoes.

### 2.6. Regulation of SlWRKY80 Expression by Exogenous SA and JA

Motelle tomato was treated with SA and MeJA at various concentration gradients, and leaves from different time points were selected for RT–qPCR analysis (Figure 7). The results showed that the expression of *SlWRKY80* increased after treatment with both hormones, and a rapid response was observed after six hours of treatment. After SA treatment at concentrations of 0.1 and 0.5 mmol·L^−1^, the dynamic expression of *SlWRKY80* first decreased and then increased, whereas its expression remained upregulated at concentrations of 2.0 and 4.0 mmol·L^−1^. At 24 h after SA treatment with 2.0 mmol·L^−1^, the expression level of *SlWRKY80* reached its maximum value (5.77-fold). *SlWRKY80* showed a bimodal expression pattern at 50, 100 and 200 μmol·L^−1^ MeJA, in which the peak expression level (3.87-fold) was observed at 18 h after 200 μmol·L^−1^ MeJA treatment. In conclusion, the expression of *SlWRKY80* was regulated by the accumulation levels of SA and MeJA, which may mediate the occurrence of a series of defense responses.

### 2.7. Subcellular Colocalization and Resistance Function Analysis of SlWRKY80

The recombinant vector *pCAMBIA2301-GFP-SlWRKY80* and the 35S-H2B-mCherry nuclear marker were transformed into *A. tumefaciens strain* GV3101, and the bacteria were injected into *N. benthamiana* at a ratio of 1:1 for subcellular colocalization analysis. SlWRKY80 was found to localize to the nucleus (Figure 8). The VIGS assay was used to silence *SlWRKY80* in Motelle tomato, and a total of 10 silenced plants showing *SlWRKY80* expression decreases of more than 50% were obtained and inoculated with *M. incognita* to evaluate disease resistance (Figure 9). The results showed that after *SlWRKY80* silencing, the number of egg masses in the roots of silenced individuals increased significantly compared with the control. The root-knot index (RKI) increased from 0.8 to 2.3, and the disease index (DI) increased from 2.2 to 46.2, showing significant changes. The final plant resistance level (RL) changed from high resistance (HR) to medium resistance (MR), reflecting the increased susceptibility of silenced lines, indicating that *SlWRKY80* could positively regulate tomato resistance to *M. incognita*.

## 3. Discussion

During the growth and development of tomatoes, plants are affected by various diseases. WRKYs often act as activators or inhibitors to participate in the defense response of tomatoes by mediating different metabolic and signal transduction pathways [20]. In the early stage of this study, soil temperature treatment was applied to individuals of the wild species LA3858 (*Mi-3*/*Mi-3*) to induce differences in susceptibility, and RNA-seq was performed after inoculation with *M. incognita*. According to the analysis of differential expression levels, six DEGs encoding WRKY proteins were finally screened as research targets.

By homologous recombinant cloning and sequencing comparison, six DEGs encoding SlWRKY1, SlWRKY13, SlWRKY30, SlWRKY41, SlWRKY46, and SlWRKY80, were identified in this study. Based on the analysis of amino acid domain differences [21], SlWKRY1, SlWKRY13, and SlWKRY46 were classified as type II WRKY proteins, while SlWKRY30, SlWKRY41, and SlWKRY80 were classified as type III WRKY proteins. The difference in grouping reflects the similarity and differences in the function of WRKY proteins, which was also reflected in the phylogenetic tree analysis. In a previous study, *AtWRKY1* was shown to regulate stomatal movement under drought stress, integrate cellular nitrogen and light energy resources, and regulate salt tolerance as a negative feedback factor in *A. thaliana* [22,23,24]. *AtWRKY13* plays a major role in the stem development of *A. thaliana*. At the same time, AtWRKY13 can interact with SPL10 to regulate *miR172b* expression, thus affecting short-day-mediated flowering. *AtWRKY13* can also activate *PDR8* expression to positively regulate cadmium tolerance [25,26,27]. *AtWRKY40* acts downstream of e-2-hexenal in *A. thaliana*, affecting the plant’s sensitivity to volatile substances. *AtWRKY40* often acts as a negative regulator in response to abscisic acid (ABA) pathways [28,29]. In this study, phylogenetic analysis revealed high homology between the above AtWRKYs and SlWRKY1, SlWRKY13, and SlWRKY46, which are very likely to be involved in plant growth, development, and stress regulation. SlWKRY30, SlWKRY41, and SlWKRY80 were divided into a single group through phylogenetic analysis, within which SlWKRY30 and SlWKRY41 were highly homologous. However, SlWRKY80 and AtWRKY70, as well as AtWRKY54, exhibited high homology. Previous studies have shown that AtWRKY70 not only participates in the activation of NB-LRR proteins by snc2-1D to induce immune responses in plants [30] but can also be ubiquitinated and degraded by CHYR1, thereby affecting immune balance [31]. In addition, WRKY70 and other WRKY proteins (WRKY46, WRKY53 or WRKY11) can synergistically mediate various signaling pathways to positively regulate the basic resistance of *A. thaliana* to *Pseudomonas syringae* [32]. AtWRKY54 can also form a protein complex with AtWRKY70 to positively regulate the expression of *SARD1* and *CBP60g* in plant immunity to resist pathogen infection [33,34]. In conclusion, *AtWRKY70* and *AtWRKY54* have important functions in plant immune processes, and *SlWRKY80*, identified in this study, is highly homologous to these *A. thaliana* genes, indicating that it is likely to have similar functions in disease resistance.

Tissue expression analysis of *SlWRKYs* showed that *SlWRKY30* and *SlWRKY80* presented higher levels of expression in roots and leaves than other *SlWRKYs*, especially *SlWRKY80*, whose accumulation in roots reached a value of 5.89-fold. The verification of the root expression pattern again demonstrated that *SlWKRY30* and *SlWKRY80* showed upregulated expression in the three dpi stage after the inoculation of resistant Motelle tomato with *M. incognita*. In the 6 dpi stage, the expression level of *SlWKRY30* decreased, while *SlWKRY80* continued to show high accumulation, reaching 4.56-fold. In conclusion, *SlWKRY80* likely plays an important regulatory role in the *Mi-1*-mediated resistance pathway.

SA and JA signaling pathways can induce the production of SAR and ISR defense mechanisms in plants, and WRKYs are often involved in regulation as vital positive or negative regulatory factors [35]. Exogenous spraying of SA and JA increased the transcription of *CaWRKY27*. The VIGS strategy demonstrated that *CaWRKY27* played a positive role in regulating the SA/JA-mediated signaling pathway involved in tobacco resistance to bacterial wilt infection [36]. Hormone treatment revealed that *VqWRKY52* was strongly induced by SA but not JA, and ectopic expression of this gene in *A. thaliana* increased resistance to powdery mildew and *Pseudomonas syringae* [37]. In tomato, an RNAi assay of the *ShWRKY81* gene in the wild species LA1777 decreased the expression of *PR* genes in the defense hormone SA pathway, resulting in increased susceptibility to powdery mildew [38]. As *SlWRKY46* is a negative regulator, its overexpression inhibited the expression of SA and JA pathway marker genes and genes encoding a pathogenesis-related protein (*PR1*) and protease inhibitors (*PI Ⅰ* and *PI Ⅱ*), which increased the susceptibility of tomato to *B. cinerea* [15]. In this work, a follow-up study of *SlWRKY80* showed that it was a typical nuclear localization protein, and VIGS functional verification showed that after silencing *SlWRKY80* in resistant Motelle tomato, the number of egg masses in the roots of individual plants significantly increased. Although the silenced plants were still resistant to disease, the resistance level mediated by *Mi-1* was reduced from HR to R, indicating that *SlWRKY80* was a positive regulator affecting *Mi-1*-mediated resistance. SA and JA accumulated rapidly in Motelle after inoculation with *M. incognita*, indicating that both hormones play an important role in *Mi-1*-mediated disease resistance. The induction rate of JA signaling was higher than that of SA signaling, which is consistent with a previous study indicating that JA is the main hormone contributing to defense against RKNs in tomatoes [39]. Exogenous SA and JA hormone treatments significantly induced the expression of *SlWRKY80*, but SA treatment obviously had a more significant effect on the upregulation of *SlWRKY80* expression. Whether *SlWRKY80* participates in defense of RKNs through more complex signal-crossing pathways needs to be further studied. In summary, the in-depth study of *SlWRKY80* will have high utility for better understanding the regulatory mechanism of WRKYs in disease resistance and improving the resistance of tomatoes to RKNs.

## 4. Materials and Methods

### 4.1. Plant Materials and Treatments

The resistant variety Motelle (*Mi-1*/*Mi-1*) and the susceptible varieties M82 and Moneymaker were used in this study. After seeds of the above materials were sown in the seedling substrate for 20–25 d, the seedlings reached a height of 12–15 cm (the fourth true leaf stage), and the seedlings were transplanted into transparent plastic nutrient pots with a diameter of 10 cm. The seedling substrate was composed of turfy soil, soil (sterilized in an incubator at 120 °C for 90 min in advance), perlite, and vermiculite at a ratio of 2:2:1:1.

### 4.2. Identification of Conserved Domains and Phylogenetic Tree Construction

The amino acid sequences of the target *SlWRKYs* were compared using Cluster 2.1 software, and the MEME online website (https://memesuite.org/meme/tools/meme (accessed on 12 March 2022)) was then used to screen highly conserved motifs and comprehensively identify conserved domains. PlantTFDB (http://planttfdb.cbi.pku.edu.cn/ (accessed on 27 March 2022)) was used to obtain WRKY proteins from *Oryza sativa* L. and *A. thaliana* that were highly homologous to each target SlWRKY, and MEGA 11 software was used to construct a phylogenetic tree with the neighbor-joining (NJ) method. The following parameter settings were applied: bootstrap value 500, part deletion = 50%, genetic distance model calculated by P-distance, and defaults for other parameter values.

### 4.3. Nematode Assay and Disease Resistance Statistics

Diseased tomato roots with root knots were washed with 1–3% NaClO for 2–3 min, followed by a running wash with distilled water for 15 min. Mature egg masses were selected from the cleaned diseased roots and placed in a sterilized petri dish containing an appropriate amount of distilled water; the number of egg masses in each dish was approximately 100–150. Finally, the egg masses were sealed in a 28 °C incubator and cultured away from light. After three days of culture, the nematode suspension was extracted, and the concentration was calculated. According to the experimental requirements, each tomato plant was inoculated with approximately 1000 J2 nematodes. According to the calculated disease index, the resistance levels of tomato materials inoculated with nematodes were classified as follows: immunity (I): DI = 0; high resistance (HR): 0 < DI ≤ 20; medium resistance (MR): 20 < DI ≤ 40; resistance (R): 40 < DI ≤ 60; sensitivity (S): 60 < DI ≤ 80; and high sensitivity (HS): 80 < DI [40].

### 4.4. Exogenous Hormone SA and JA Treatment

SA and JA were prepared at concentrations of 0.1, 0.5, 1, 2 and 4 mmol·L^−1^ with distilled water, acetone, and 0.1% Tween 80 (*v*/*v*). After cultivation, Motelle tomato plants were sprayed with hormones at the four-true leaf stage. Untreated leaves and leaves collected 6, 12, 18 and 24 h after treatment were selected to evaluate the accumulation of *SlWRKY80*.

### 4.5. Determination of Endogenous SA and JA

Leaves were sampled from Motelle and M82 seedlings, and the JA and SA in the samples were separately extracted and purified, with three biological replicates per time point. Then, all the samples were analyzed using an HPLC electrospray ionization/MS-MS system (Alliance HPLC 1525, Waters, MA, USA).

The parameters for SA level determination were as follows: Waters 2695 chromatograph; column type, Elite SinoChrom C18 (4.6 mm × 150 mm, 5 μm) column; mobile phase, methanol; sodium acetate (0.2 mol/L, pH 5.5) volume ratio, 6:4 mixture; injection volume, 10 μL; and flow rate, 0.5 mL/min. The parameters for JA level determination were as follows: column temperature, 50 °C; mobile phase A, water (containing 0.001% formic acid); mobile phase B, acetonitrile; flow rate, 0.2 mL/min; gradient elution: 80% A + 20% B → 10% A + 90% B (0~8 min), 10% A + 90% B (8.1~10 min), 80% A + 20% B (10.1~12.0 min); and injection volume, 5 μL.

### 4.6. Expression Pattern Analysis

RT–qPCR was performed on qTOWER3G (Analytik Jena AG, Jena, Germany). Two microliters of cDNA were used as a template for amplification in each reaction in each run. The 20 μL system contained 10 μL of Vazyme ChamQ SYBR qPCR Master Mix, 0.4 μL forward primer, 0.4 reverse primer and 7.2 μL of ddH_2_O. The reaction procedure was as follows: denaturing at 95 °C for 30 s and amplification for 40 cycles of 95 °C for 10 s and 60 °C for 30 s. The temperature range was 95 °C–60 °C–95 °C for 15 s, 60 s and 15 s, respectively, for melting curve analysis. Three independent experimental replications were performed. The ΔΔCt calculation method was performed following a previous study [41], in which *PIF1* and *ACT3* were used as reference genes in root expression pattern and tissue expression analysis, respectively.

### 4.7. Screening and Homologous Recombinant Cloning of SlWRKYs

According to the RNA-seq results of variety LA3858 (*Mi-3*/*Mi-3*), differentially expressed (log_2_FC ≥ 1) WRKYs were screened from the anti-sensitivity group. Finally, six genes, *Solyc12g006170.2*, *Solyc01g095630.3*, *Solyc03g095770.3*, *Solyc04g051540.3*, *Solyc08g067340.4*, and *Solyc10g009550.3*, were selected for homologous recombinant cloning. The cDNAs of the target *SlWRKYs* were obtained from the SGN database (https://solgenomics.net/ (accessed on 13 May 2021)). Then, total RNA was extracted from the leaves of the Moneymaker variety using a kit (Vazyme, RC401) and reverse-transcribed into cDNA with a kit (Vazyme, R211-01). Specific upstream and downstream primers with 15 bp recombinant homologous arms were designed with *Bam* HⅠ as the digestion site, and the CDSs were amplified by PCR (Appendix A). After amplification, the fragments were purified, recovered and inserted into plasmid *pBI121* (Appendix A), and the recombinant vector was transformed into *E. coli* DH5α and cultured on solid LB medium (containing 50 μg·mL^−1^ kanamycin) for 12 h away from light. Monoclonal colonies were selected and cultured at 37 °C and 220 r·min^−1^ for 13 h. Subsequently, PCR was performed to identify a positive clone for sequencing.

### 4.8. VIGS Experiment

The *pTRV2* plasmid was selected as the target fragment vector, and the CDS region of *SlWRKY80* was amplified by PCR using specific primers with 15 bp homologous arms. *pTRV2* was digested by *Eco* RⅠ, and the silencing expression vector *pTRV2-SlWRKY80* was then constructed (Appendix A). *A. tumefaciens* strain GV3101 was cultured on LB solid medium (containing 50 μg·mL^−1^ kanamycin) with 10 μL of the recombinant vector at 28 °C for 36 h, followed by single-colony selection. Single colonies were selected and placed in 3 mL liquid LB containing antibiotics (50 μg·mL^−1^ kanamycin and 25 μg·mL^−1^ rifampin) and then cultured at 30 °C and 220 r·min^−1^ for 18 h. The culture solution of strain GV3101 was poured into 20 mL liquid LB (containing 50 μg·mL^−1^ kanamycin and 25 μg·mL^−1^ rifampin), followed by incubation for 18 h. The expanded *A. tumefaciens* cells were centrifuged, the supernatant was discarded, and the O.D. value was determined and adjusted to 0.3 before use.

The *pTRV1* bacterial solution was mixed 1:1 with *pTRV2-PDS*, *pTRV2* no-load and *pTRV2-SlWRKY80* bacterial solutions. The mixed bacterial solution was placed at 4 °C overnight, and the plants were infected by vacuum extraction. After 72 h of cultivation under darkness and high humidity, normal plant management conditions of 14 h light/10 h dark cycles, appropriate temperatures of 16–25 °C, and 60% humidity were applied.

### 4.9. Subcellular Localization of SlWRKY80

The CDS region of *SlWRKY80* was amplified, and a *pCAMBIA-GFP-SlWRKY80* recombinant vector was constructed (Appendix A). The constructed plasmid was transferred into *A. tumefaciens strain* GV3101, and the bacteria were cultured at 30 °C for 2 d. Monoclonal *A. tumefaciens* was selected for suspension and centrifuged at 4000 rpm for 4 min, the supernatant was removed, and bacteria were collected. The bacteria were injected with an infection solution (10 mM MgCl_2_ + 10 mM Mes + 20 μM acetoeugenone), the OD600 was adjusted to approximately 0.8, and the bacteria were activated at 28 °C for 2 h. After the bacterial solution was resuspended and placed at 28 °C for 2 h, the leaves of *N. benthamiana* were injected, and the treated individuals were cultured for 2 d in low light. The results were observed by confocal laser microscopy (Olympus FV1000); the 35S-H2B-mCherry fusion protein was used as a positive control.

## Figures and Tables

**Figure 1 plants-12-02416-f001:**
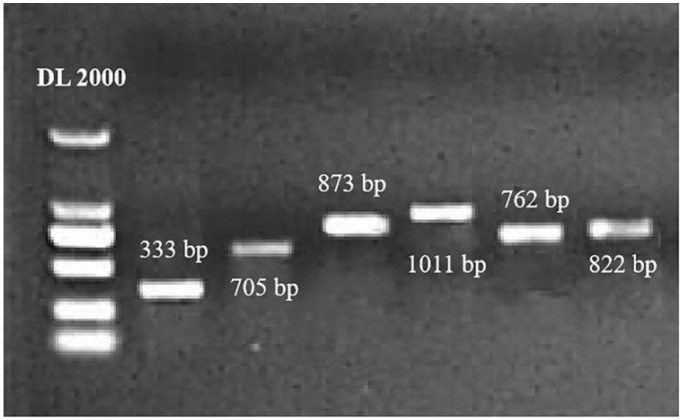
PCR detection of *SlWRKY*-positive clones. Lane 1 represents the DL 2000 marker, and lanes 2–7 represent the CDS band sizes of the target genes *SlWRKY1*, *SlWRKY13*, *SlWRKY30*, *SlWRKY41*, *SlWRKY46,* and *SlWRKY80*, respectively.

**Figure 2 plants-12-02416-f002:**
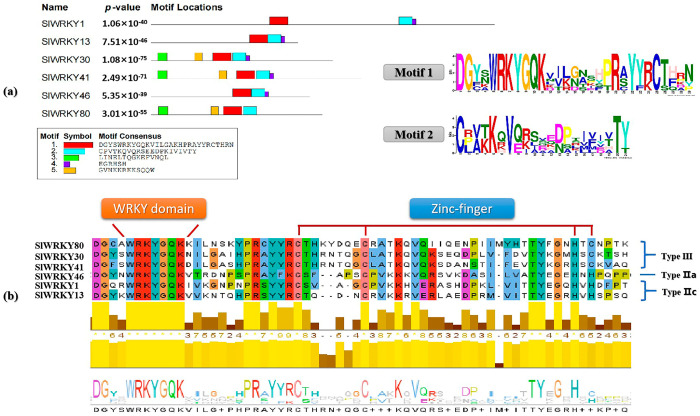
Identification and classification of conserved SlWRKY domains. (**a**) Screening of highly conserved motifs in the SlWRKY domain; (**b**) Classification according to the conserved structure of SlWRKYs. According to conservation scores, five highly conserved motifs were screened, which determined the different functions of SlWRKYs due to their different numbers and locations in the structural domains, resulting in inconsistent SlWRKY classifications.

**Figure 3 plants-12-02416-f003:**
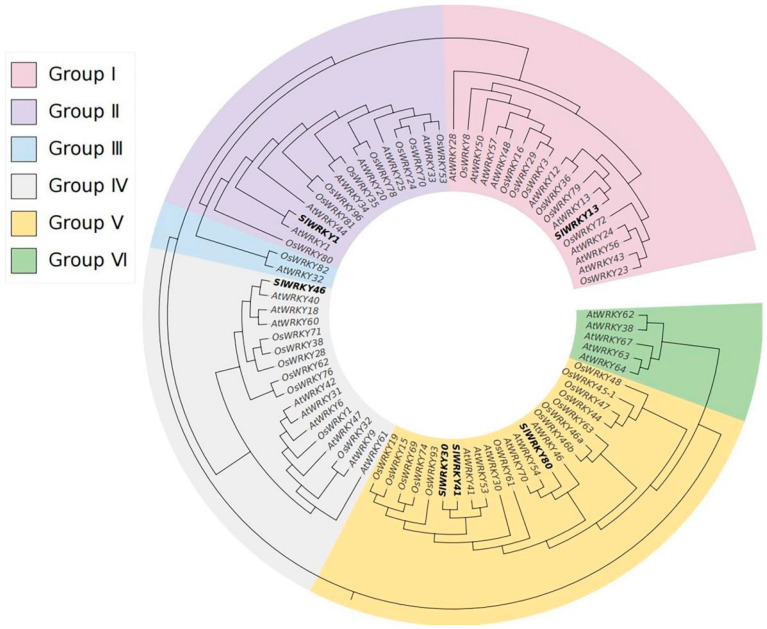
Phylogenetic tree analysis of WRKYs. The phylogenetic tree was constructed by obtaining WRKY proteins (36 from *A. thaliana* and 36 from *Oryza sativa* L.) with good homology to the six SlWRKYs identified in this study.

**Figure 4 plants-12-02416-f004:**
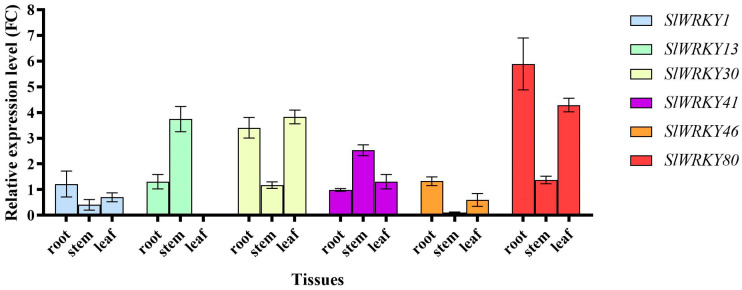
Determination of *SlWRKY* accumulation levels in different tissues of Motelle tomato. Different root, stem, and leaf tissues were selected from the four-true leaf stage of Motelle individuals, and an RT–qPCR strategy provided guidance to verify the expression levels of *SlWRKYs*.

**Figure 5 plants-12-02416-f005:**
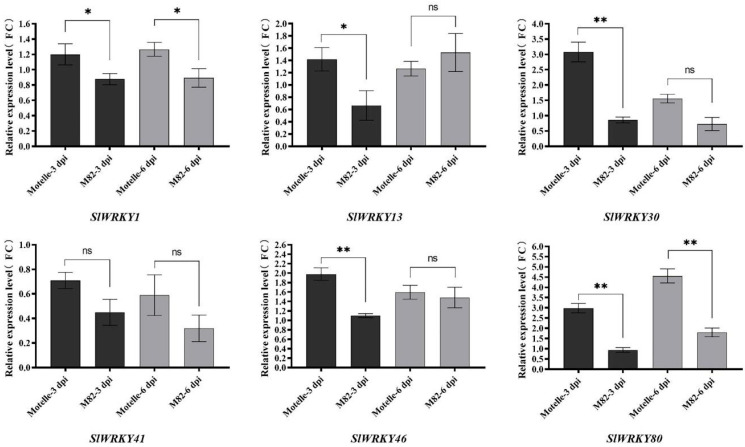
Expression of *SlWRKYs* at three dpi and six dpi in Motelle and M82 tomatoes. The RT–qPCR strategy provided guidance for verifying the expression levels of *SlWRKYs*, “**”, “*” and “ns” indicate significant differences relative to each control according to a *p*-value < 0.05 based on one-way ANOVA. Each value was the mean ± SD of three biological determinations.

**Figure 6 plants-12-02416-f006:**
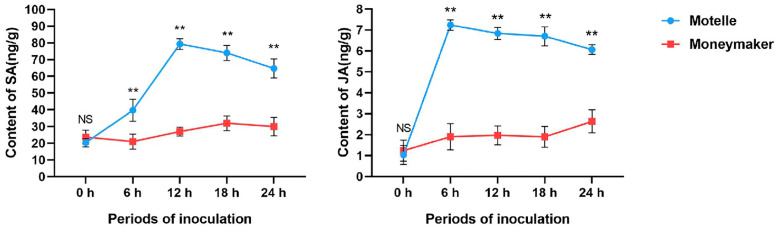
Determination of the contents of the endogenous hormones SA and JA in resistant tomato Motelle and susceptible tomato M82. Untreated and treated individuals were selected at 6, 12, 18 and 24 hours (h) for the determination of SA and JA with HPLC-MS/MS. “**” and “ns” represent the significance of differences relative to each control, according to a *p*-value < 0.05, based on one-way ANOVA. Each value is the mean ± SD of three biological replicates.

**Figure 7 plants-12-02416-f007:**
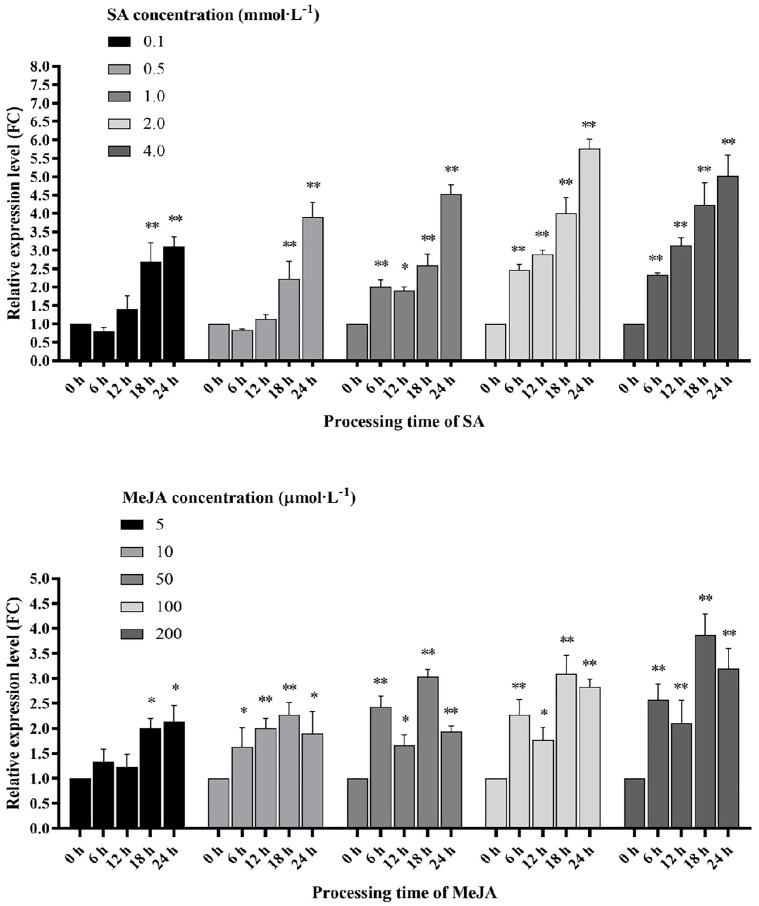
Effects of spraying SA and JA at different concentrations on *SlWRKY80* expression patterns. Different concentrations of SA and JA were used for spraying, and untreated and treated leaves were collected at 6, 12, 18 and 24 h for the determination of *SlWRKY80* expression statistics. The RT–qPCR strategy provided guidance for verifying the expression levels of *SlWRKY80*, “**” and “*” represent significant differences relative to each control, according to a *p*-value < 0.05, based on one-way ANOVA. Each value is the mean ± SD of three biological replicates.

**Figure 8 plants-12-02416-f008:**
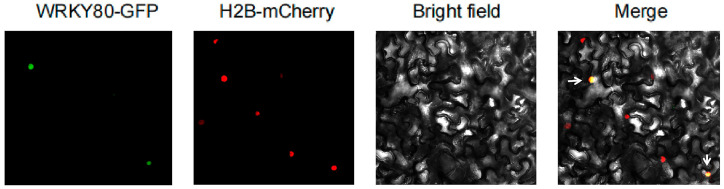
Subcellular localization of SlWRKY80. Constitutive GFP served as a control, while 35S-H2B-mCherry was used as a nuclear localization marker. The constructs were transiently expressed in *N. benthamiana* leaves. The green signal of GFP merged with the red signal of the nuclear marker to form an orange signal (indicated by the white arrow in the figure), indicating the localization of the protein. Green fluorescent protein (GFP): excitation wavelength was 488 nm, and emission wavelength 510 nm.

**Figure 9 plants-12-02416-f009:**
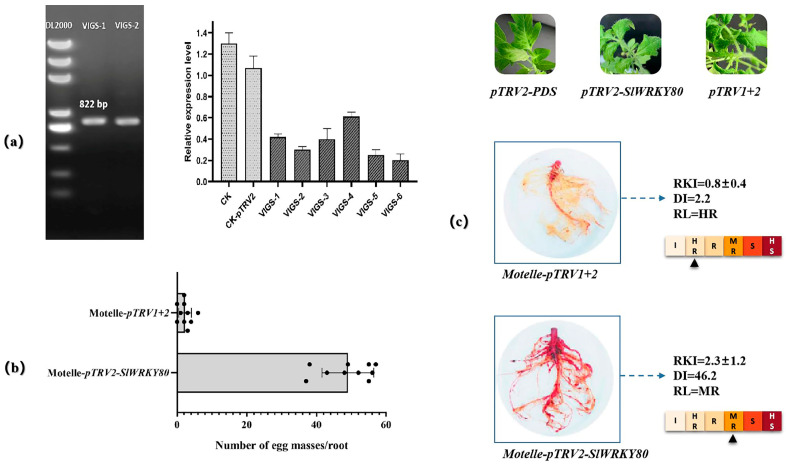
Functional identification of *SlWRKY80* was conducted via the VIGS strategy. (**a**) *pTRV-SlWRKY80*-positive clone screening and plant silencing efficiency verification. (**b**) Statistical comparison of the number of egg masses between *SlWRKY80*-silenced plants and CK plants. (**c**) Statistics of changes in the RKI, DI, and RL values of *SlWRKY80*-silenced and CK plants. The RT–qPCR strategy provided guidance for verifying the expression level of *SlWRKY80* based on one-way ANOVA. Each value was the mean ± SD of three biological determinations.

## Data Availability

Not applicable.

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
