# Peer review of "Identification of the Regulatory Role of SlWRKYs in Tomato Defense against Meloidogyne incognita"

_plants, 2023, doi:10.3390/plants12132416_

Round 1

Reviewer 1 Report

In the manuscript entitled “Identification of the regulatory role of SlWRKYs in tomato defense against Meloidogyne incognita”, the authors used homologous recombinant cloning and sequencing comparison to identify six genes that were differentially expressed in wild-type tomato species LA3858 (Mi-3/Mi-3) at different soil temperatures and after inoculation with the root-rot nematode, Meloidogyne incognita. They determined that the six genes encode WRKY transcription factors, SlWRKY1, SlWRKY13, SlWRKY30, SlWRKY41, SlWRKY46 and SlWRKY80. Severe root-rot nematode (RKN) infestation causes dramatic yield loss in several crops, resulting in devastating economic losses to farmers and producers. Presently, the most effective method of mitigating the effects of RKN on crops is to identify RKN resistant (R) genes and manipulate these genes to develop resistant RKN varieties. Currently, while tomato is highly susceptible to RKN infestation, the genes related to RKN resistance are in wild tomato species. Thus, there are limitations to their use in cultivated varieties because of incompatibility.  However, resistance to plant pathogens is also regulated by several other biological factors, including the WRKY transcription factors (TFs).  

WRKY TFs have been shown to regulate pattern-triggered immunity (PTI) and effector-triggered immunity (ETI) pathways. Some also function in salicylic acid (SA) and jasmonic acid (JA) defense pathways, and even play a role in regulating expression of R genes. Previous studies have indicated that there are over 80 WRKY TFs in tomato (SlWRKYs). Of these SlWRKYs, only a small group, including SlWRKY72, SlWRKY73, SlWRKY74, SlWRKY45, SlWRKY3 and SlWRKY35, have been shown to regulate resistance to RKNs. Therefore, the role of the other SlWRKYs in tomato disease resistance is yet to be explored. In this study, the authors investigated differentially expressed SlWRKY1, SlWRKY13, SlWRKY30, SlWRKY41, SlWRKY46 and SlWRKY80 to determine how they regulate tomato resistance to RKN. Phylogenetic analysis revealed that SlWRKY1, SlWRKY13 and SlWRKY46 are functionally similar and they play a role in abiotic stress responses and growth and development, while SlWKRY30 and SlWKRY41 is highly homologous. SlWRKY80 shares high sequence homology with Arabidopsis WRKYs (AtWRKY54 and AtWRKY70) which function in plant immune processes. SlWRKY80 was highly expressed in the roots and leaves of the RKN resistant variety Motelle (Mi-1/Mi-1) and was continuously upregulated in the roots of Motelle after inoculation with Meloidogyne incognita. This data suggested that SlWRKY80 is involved in Mi-1-mediated disease resistance. While VIGS assay confirmed the involvement of SlWRKY80 in tomato resistance to RKNs, assays with SA and JA treatments showed that SlWRKY80 responded rapidly, indicating that SlWRKY80 may be regulating Mi-1-mediated disease resistance through hormone signaling.

 Is the manuscript clear, relevant for the field and presented in a well-structured manner? 

Yes, the manuscript is clear, relevant for the field and presented in a well-structured manner.

Are the cited references mostly recent publications (within the last 5 years) and relevant? Does it include an excessive number of self-citations?

              Yes, the references are recent and relevant, without excessive self-citations.

Is the manuscript scientifically sound and is the experimental design appropriate to test the hypothesis?

Yes, the manuscript is scientifically sound and the experimental design is appropriate to test the hypothesis.

Are the manuscript’s results reproducible based on the details given in the methods section?

Yes, the methods are detailed and can be easily followed to repeat the results presented.

Are the figures/tables/images/schemes appropriate? Do they properly show the data? Are they easy to interpret and understand? Is the data interpreted appropriately and consistently throughout the manuscript?

Yes, all of the figures are appropriate, simple to follow and to understand. However, for figure 7, it is difficult to see the overlap of the merged GFP with mCherry. The authors might consider including a better image and using arrow heads to highlight the merged markers.

Are the conclusions consistent with the evidence and arguments presented?

Yes, the results support the conclusions drawn.

 1.      What is the main question addressed by the research?

To date, only a small group of SlWRKY TFs have been shown to regulate resistance to root-rot nematode (RKN), Meloidogyne incognita in tomato. Therefore, the role of the other SlWRKYs in tomato disease resistance is yet to be explored.

 In this study, the authors wanted to investigate the role of additional SlWRKY TFs in resistance to RKN in tomato. They used homologous recombinant cloning and sequencing comparison to identify six WRKY genes that were differentially expressed in wild-type tomato species LA3858 (Mi-3/Mi-3) at different soil temperatures and after inoculation with the root-rot nematode, Meloidogyne incognita. The authors also investigated the role of these genes in tomato resistance to RKN.

 2. Do you consider the topic original or relevant in the field?

This study is relevant to the field because, for the first time, the role of SlWRKY1, SlWRKY13, SlWRKY30, SlWRKY41, SlWRKY46 and SlWRKY80 in tomato disease resistance is explored.

 Does it address a specific gap in the field?

For the first time, the authors show that SlWRKY1, SlWRKY13 and SlWRKY46 are functionally similar and that they play a role in abiotic stress responses and growth and development, while SlWKRY30 and SlWKRY41 are highly homologous. SlWRKY80 shares high sequence homology with Arabidopsis WRKYs (AtWRKY54 and AtWRKY70) which function in plant immune processes. SlWRKY80 was highly expressed in the roots and leaves of the RKN resistant variety Motelle (Mi-1/Mi-1) and was continuously upregulated in the roots of Motelle after inoculation with Meloidogyne incognita. This data suggested that SlWRKY80 is involved in Mi-1-mediated disease resistance.

 3. What does it add to the subject area compared with other published material?

The results presented in this study show for the first time the involvement of SlWRKY80 in tomato resistance to RKNs, and that SlWRKY80 is regulating this resistance through hormone signaling.

 4. What specific improvements should the authors consider regarding the methodology? What further controls should be considered?

The methodology is straightforward and nothing significant is lacking to replicate the analyses and experiments.

No additional controls are needed for the scope of this study.

 5. Are the conclusions consistent with the evidence and arguments presented and do they address the main question posed?

Yes, the conclusions are consistent with the results provided. All of the data presented are relevant to the main question the authors are addressing.

 6. Are the references appropriate?

Yes, the references are appropriately relevant to the study.

 7. Please include any additional comments on the tables and figures.

In figure 7, it is difficult to see the overlap of the merged GFP with mCherry. The authors might consider including a better image and using arrow heads to highlight the merged markers.

 There are also several typing errors throughout the manuscript that should be fixed before publication. For example:

line 10-11- WRKY transfection factors are involved in complex, diverse biological processes in plants. That should be “transcription factors”….

 Line 246-248 - In this study, phylogenetic analysis revealed high homology between the above AtWRKYs and SlWRKY1, SlWRKY13 and SlWRKY40, which are very likely to be involved in plant growth, development and stress regulation. That should be “SlWRKY46”….

There are no major English issues in this manuscript. However, there are a few typing mistakes that should be fixed prior to publication. 

Author Response

Dear Editors and Reviewers:

Thank you for your letter and for the Reviewers’ comments concerning our manuscript entitled “Identification of the regulatory role of SlWRKYs in tomato defense against Meloidogyne incognita”. (ID: plants-2444665). Those comments are all valuable and very helpful for revising and improving our paper, as well as the important guiding significance to our researches. We have studied comments carefully and have made correction which we hope meet with approval. Revised portions are marked with "track changes" in the revised manuscript. The main corrections in the manuscript and the responds to the Reviewer's comments are as follows:

  1. According to the requirements of the reviewers, we have deeply modified the language of the manuscript, and provided AJE editing service proof.
  2. It is really true as Reviewer suggested that In original figure 7, it is difficult to see the overlap of the merged GFP with mCherry. Therefore, we have made relevant adjustments to the clarity and observation magnification of the figure (Figure 8 in revised manuscript), and added indicating arrows to facilitate the reader to accurately capture the figure information.
  3. We are very sorry for our incorrect writing and description errors in the manuscript. For example, the Reviewer identified "transfection factors" in lines 10-11 of the original manuscript that were modified in the same lines of the revised manuscript. The "SlWRKY40" in line 246-248 of the original manuscript have been correctly modified to "SlWRKY46" in line 270 of the revised manuscript. Then we carried out a detailed examination and correction of the typing errors throughout the whole manuscript.

Finally, we would like to express our gratitude to the Reviewer for recognition of the value and significance of this study, as well as for the valuable feedbacks on the manuscript.

Reviewer 2 Report

The article “Identification of the regulatory role of SlWRKYs in tomato defense against Meloidogyne incognita” is an interesting compilation of data and has merits for publication. However, in its current form, the article contains major flaws and errors that must be revised before publication. Below are my suggestions which I believe will improve the impact of this article.

Major comments

First things first, the language of the manuscript needs to be improved thoroughly. Some statements are hard to read.

The abstract is written poorly. I would strongly suggest rewriting it.

In material and method (lines 332-336), why were the hormones sprayed on leaves but not on roots? Also, following hormonal application, the expression of SlWRKY80 was investigated in leaves but not in roots. Since the RKN infection occurs in roots, applying hormone or investigating the expression on/in leaf is unnecessary. This is a major problem, and the author needs to provide an explanation or redo the experiment.

The quantification of SA and MeJA is essential in Motelle (Mi-1/Mi-1), M82, and Moneymaker. I strongly suggest doing a hormonal quantification assay of root tissue collected from Motelle (Mi-1/Mi-1), M82, and Moneymaker.

The expression of SlWRKY80 induced by SA and MeJA. It would be interesting to see the binding activity of SlWRKY80 with JA and SA biosynthesis genes using Y1H and LUC assay.

Since the author centered his/her focus on SA and MeJA hormones, the expression level of defense-related SA and MeJA genes should be performed in WT and pTRV2-SlWRKY80 plants.

Why did the author ignore cytokinin? Previous studies suggested a compelling role of cytokinin in regulating plant response to RKN.

Minor comments 

The author performed expression analysis of SlWRKY80 in different tissues (Figure 4). It is tough to understand owing to the similarities of histogram color. I would strongly suggest making amendments.

The quality of graphs is below substandard. I would suggest colorizing the graphs for better readability.

Reference number 6 (line 423-424) can be updated with Involvement of auxin in growth and stress response of cucumber (maxapress.com) doi: 10.48130/VR-2022-0013. 

Moderate editing of English language required.

Author Response

Dear Editors and Reviewers:

Thank you for your letter and for the Reviewers’ comments concerning our manuscript entitled “Identification of the regulatory role of SlWRKYs in tomato defense against Meloidogyne incognita”. (ID: plants-2444665). Those comments are all valuable and very helpful for revising and improving our paper, as well as the important guiding significance to our researches. We have studied comments carefully and have made correction which we hope meet with approval. Revised portions are marked with "track changes" in the revised manuscript. The main corrections in the manuscript and the responds to the Reviewer's comments are as follows:

  1. According to the requirements of the Reviewer, we have deeply modified the language of the manuscript, rewriten the abstract, and provided AJE editing service proof.
  2. As we know, systemic resistance mechanisms in plants include SAR and ISR, which depend on SA and JA signals, respectively. In this study, we also found that SlWRKY80 did not exhibit tissue-specific expression in the roots and leaves of the resistant variety Motelle. The purpose of this study was only to observe the effects of SA and JA on the accumulation level of SlWRKY80, and to determine whether SlWRKY80 regulates plant disease resistance by participating in SA and JA signaling pathways. Therefore, we chose to use the more efficient method, which is to deal with the leaves. However, the hormone treatment of root and the exploration of SlWRKY expression in root proposed by Reviewer are more targeted and have high guiding significance for the improvement of our later research. So we're going to be doing this series of explorations later on.
  3. It is really true as Reviewer suggested that the quantification of SA and MeJA is essential in Motelle (Mi-1/Mi-1) and M82. Through the accumulation levels of SA and JA in disease-resistant and susceptible varieties, the role of both signals in the Mi-1-mediated disease resistance pathway was clarified, which provided a more perfect logic for further exploration of the regulatory role of SlWRKY80 through the two signals. We have added previous experimental data to the revised manuscript, line 177-197, figure 6.
  4. In this study, we identified the high regulatory value of SlWRKY80 in the process of tomato defense against root knot nematode through protomic data, bioinformatics analysis, hormone tests and VIGS. As suggested by the Reviewer, we will screen interaction targets through Y1H, LUC or EMSA in the later stage, combined with protein level verification including Co-IP and GST Pull down, explore the key genes in SA and JA signals affected by SlWRKY80, and clarify the regulatory mechanism of SlWRKY80. Finally, it was used in the resistance improvement of high quality tomato germplasm. Thanks again to the Reviewer for this valuable advice. 
  5. We have not involved in the study of cytokinin on the regulation of plants response to nematodes, and we will pay more attention to this research in the future, thanks to the Reviewer for this valuable suggestion.
  6. Considering the Reviewer's suggestion, we have changed the histogram color of figure 4 so as to make the reader get the picture information more clearly.
  7. As Reviewer suggested that we check the clarity of the graphs again and ensure that each has a resolution of 300 dpi or higher.
  8. We have updated reference 6 in lines 460-461 according to the Reviewer's comments.

Finally, we would like to express our gratitude to Reviewer for the valuable and forward-looking suggestions, which provide important guidance for both the improvement of manuscript quality and the improvement of our team's scientific research level.

Reviewer 3 Report

This manuscript meets the Scope of the  Plant Journal. The study is interesting and results are valuable. However, I have several minor points: 

Line 79: please add a space after M. javanica[19].

Line 202: please add a space after evaluate disease resistance(Figure 8).

Line 345: please add a space after previous study[41],

It would be better to show each PCR reaction with its negative control. If you have one, please show it

 When appearing for the first time, abbreviations should be explained. You should add the meaning of dpi (days post-infection)

Author Response

Dear Editors and Reviewers:

Thank you for your letter and for the Reviewers’ comments concerning our manuscript entitled “Identification of the regulatory role of SlWRKYs in tomato defense against Meloidogyne incognita”. (ID: plants-2444665). Those comments are all valuable and very helpful for revising and improving our paper, as well as the important guiding significance to our researches. We have studied comments carefully and have made correction which we hope meet with approval. Revised portions are marked with "track changes" in the revised manuscript. The main corrections in the manuscript and the responds to the Reviewer's comments are as follows:

  1. According to the requirements of the Reviewer, we have deeply modified the language of the manuscript, and provided AJE editing service proof.
  2. Considering the Reviewer's suggestion, we have added a space between "M. javanica" and "[19]" in line 81 of revised manuscript.
  3. Considering the Reviewer's suggestion, we have added a space between "resistance" and "(Figure 9)" in line 225 of revised manuscript.
  4. Considering the Reviewer's suggestion, we have added a space between "study" and "[41]" in line 382 of revised manuscript.
  5. As Reviewer suggested that we have added the negative control to the method 4.6 in lines 382-384 of revised manuscript.
  6. We have made correction according to the Reviewer's comments. The abbreviations of the full manuscript are checked and modified. For example, in lines 158-159 of the revised manuscript, the first occurrence of ”dpi“ is explained.

Finally, I would like to thank the Reviewer again for the valuable comments on our manuscript.

Round 2

Reviewer 2 Report

Accept for publishing.

Author Response

Dear Reviewer:

Thank you for your very instructive and valuable suggestions for revisions to the manuscript. In the next in-depth study of this gene, we will refer to your suggestions to find valuable targets for combined verification, so as to explore the transcriptional regulation mechanism of disease resistance.

Finally, thank you again for your guidance in this study!